# Lymphocyte subsets in Atlantic cod (*Gadus morhua*) interrogated by single-cell sequencing

Naomi Croft Guslund [1✉], Anders K. Krabberød [1,2], Simen F. Nørstebø[3], Monica Hongrø Solbakken[1], Kjetill S. Jakobsen [1], Finn-Eirik Johansen[4] & Shuo-Wang Qiao[5✉]

Atlantic Cod (*Gadus morhua*) has lost the *major histocompatibility complex class II* presentation pathway. We recently identified CD8-positive T cells, B cells, and plasma cells in cod, but further characterisation of lymphocyte subsets is needed to elucidate immune adaptations triggered by the absence of CD4-positive T lymphocytes. Here, we use single-cell RNA sequencing to examine the lymphocyte heterogeneity in Atlantic cod spleen. We describe five T cell subsets and eight B cell subsets and propose a B cell trajectory of differentiation. Notably, we identify a subpopulation of T cells that are CD8-negative. Most of the CD8-negative T lymphocytes highly express the homologue of *monocyte chemotactic protein 1b*, and another subset of CD8-negative T lymphocytes express the homologue of the scavenger receptor *m130*. Uncovering the multiple lymphocyte cell sub-clusters reveals the different immune states present within the B and T cell populations, building a foundation for further work.

[1] Centre for Ecological and Evolutionary Synthesis (CEES), Department of Biosciences and the Department of Immunology, University of Oslo, Oslo, Norway. [2] Section for Genetics and Evolutionary Biology, Department of Biosciences and the Department of Immunology, University of Oslo, Oslo, Norway. [3] Department of Paraclinical Sciences, Faculty of Veterinary Medicine, Norwegian University of Life Sciences, Ås, Norway. [4] Section for Physiology and Cell Biology, Department of Biosciences, University of Oslo, Oslo, Norway. [5] Department of Immunology, Institute of Clinical Medicine, University of Oslo, Oslo, Norway. ✉email: n.c.guslund@ibv.uio.no; s.w.qiao@medisin.uio.no

Comparative studies of a limited number of model organisms have led to a commonly held view that the adaptive immune system common to jawed vertebrates has remained largely unchanged since its origin, suggested to have arisen some 500 million years ago[1,2]. More recent studies of non-model organisms have begun to challenge this view, as exemplified by the Atlantic cod (*Gadus morhua*)[3]. The Atlantic cod belongs to a growing number of identified teleost species that have lost substantial parts of the adaptive immune system. The Gadiformes order, of which the Atlantic cod is one of the most iconic species, lost the entire set of *major histocompatibility complex (MHC) class II* genes around 80–100 million years ago[3,4]. The *cd4* gene is also non-functional, and thus cod does not have CD4-positive T cells, which are known to play a pivotal role in raising a humoral immune response in mammalian immune systems. Despite its 'unusual' immune system, cod is a prolific species and in its natural environment is not more prone to diseases than other teleosts with more 'conventional' immune systems[5]. Although it has proven difficult to elicit specific antibody responses by immunisation, Atlantic cod does display specific and long-term protective memory in vaccination studies[6–10]. Thus, Atlantic cod has a well-functioning immune system in the absence of CD4-positive T cells, but the mechanisms for generating specific immune memory, if it exists in cod, must differ from the well-known model based on collaboration between B cells and CD4-positive T cells, although this model is in general not validated in teleosts. Germinal centres, the site of mammalian B cell - CD4-positive T-cell interaction, are not present in teleosts. Affinity maturation of B cells does occur to some extent in teleosts during immune responses, as demonstrated by the replacement of low-affinity immunoglobulins (Igs) by Igs of increasing affinity in rainbow trout[11], although this appears to be less efficient in fish than in mammals[12]. Thus, while teleosts do demonstrate somatic mutation of their B-cell receptors with some degree of affinity maturation; it has not yet been proven that CD4-positive T cells are necessary for this process.

As a first step towards understanding the workings of the immune system of Atlantic cod, we have used single-cell transcriptomics to characterise the immune cell composition. In a previously published study, we analysed the transcriptomic profile of some seven thousand immune cells sampled from the spleen and blood of two Atlantic cod[13]. We identified major immune-cell populations (T cells, B cells, plasma cells, macrophages, and granulocytes) and non-immune cells (erythrocytes, thrombocytes, and spleen stromal cells), as well as minor populations of cells such as dendritic cells and GATA3+ cytotoxic cells, a name coined by us as there was no clear classification. We hypothesised that a subset of T cells might functionally resemble CD4-positive T cells, expressing markers that may indicate their interaction with B cells. However, we did not have sufficient cell numbers for further sub-clustering of the 1300 T cells.

In the current work, we have assembled a much larger dataset of single-cell RNA sequencing (scRNAseq) data of 57 thousand spleen cells from 34 Atlantic cod specimens in a vaccination study. The teleost spleen functions as the primordial secondary lymphoid organ where adaptive immune responses are generated[2]. By sampling from both naive and vaccinated fish before, during, and after an immune challenge, we capture cells in a broad spectrum of immune states. Sub-clustering and pseudo-time trajectory analysis of the B cells revealed expression of transcription factors that may be involved in B cell differentiation. Closer inspection of the almost 14 thousand T lymphocytes revealed a large subset that does not express CD8. The majority of CD8-negative T cells express high to extremely high levels of a homologue of *monocyte chemotactic protein 1b* (*mcp1b*), whereas a smaller subset of CD8-negative T cells express a homologue of the scavenger receptor *m130*. The presence of T cell subsets in the Atlantic cod secondary lymphoid organ that express neither *cd4* nor *cd8* co-receptors poses important questions about the functional organisation of the immune system in this species.

## Results

### Overview of cells found in Atlantic cod spleen

We analysed spleens from 34 Atlantic cod at 12 timepoints during a vaccination and immune challenge study of *Vibrio anguillarum*, thus capturing a broad immune status. We examined these spleen samples using scRNAseq. In this study, we focus on a detailed exploration of cell sub-clusters within the B and T cell lymphocytes, and thus we do not delve into the differing immune responses following vaccination and immune challenge. All the named clusters in the global UMAP in Fig. 1 as well as the larger lymphocyte sub-clusters have cells from all the sampled fish. While some of the smallest lymphocyte sub-clusters are not represented by all fish, there was no pattern of changing sub-cluster population size with respect to naive versus vaccinated fish or along the immune challenge timeline. This demonstrates that all the splenic cell populations and lymphocyte sub-cluster populations are found in both steady-state and immune-perturbed Atlantic cod. The number of cells found in each cluster or sub-cluster is shown in Supplementary data 4.

Following quality control filtering, we derived a gene expression matrix across 56,994 Atlantic cod splenic cells. Visualisation of cell types in two dimensions using UMAP revealed 15 cell clusters with distinct gene expression signatures (Fig. 1a, a list of differentially expressed genes can be viewed in Supplementary data 2). Three clusters, clusters 16, 17, and 18, each represented less than 0.2% of the total cells and so remain unnamed. The splenic cells contained the major types of lymphoid and myeloid cells, minor immune cell types (myeloid cell type, dendritic cell cluster 1 and 2, and GATA3+ cytotoxic cells), as well as other cell types (the thrombocytes, erythrocytes, spleen stromal cells, and endothelial cells). The cell cluster 'myeloid cell type' was previously named 'natural killer cells' by us[13]. Despite a larger sample size in the present study, a lack of clear marker genes has resulted in a less specific name choice. Two clusters demonstrated gene expression profiles matching dendritic cells, both expressing the receptor tyrosine kinase *flt3*, allograft inflammatory factor 1 (*aif1*), and various cathepsin genes, and have thus been named dendritic cell cluster 1 and 2. The naming of the cell types is otherwise consistent with our first classification of cod immune cells[13].

The lymphocytes represented 41% of the splenic cells present, and were made up of B cells, plasma cells, proliferating lymphocytes, and T cells (Fig. 1b). The T cells were clustered into two distinct groups; one group contains CD8-positive T cells, while the other did not show CD8 expression but highly expressed a homologue of *mcp1b*, thus given the name MCP1b-positive T cells (Fig. 1a). The myeloid cells (containing the macrophages, neutrophils, and myeloid type cells) represented 14% of splenic cells, while the other immune cells (Dendritic cell cluster 1, Dendritic cell cluster 2, and GATA3+ cytotoxic cells) represented 5% of splenic cells (Fig. 1b).

### B lymphocyte sub-clusters

By selecting 9474 B lymphocytic cells (the B cells, B cells within the proliferating lymphocytes, and plasma cells) from the global UMAP containing all splenic cells and re-clustering these cells at a higher resolution, eight sub-clusters were revealed (Fig. 2a). Shared expression of the marker genes *cd22*, *swap70*, *cd79a* (Fig. 2b) and immunoglobulin genes (Supplementary Fig. 3) confirmed these were all B lymphocytes. B cell sub-clusters 0, 1, 2, 3, 4, and 6 were not conspicuously

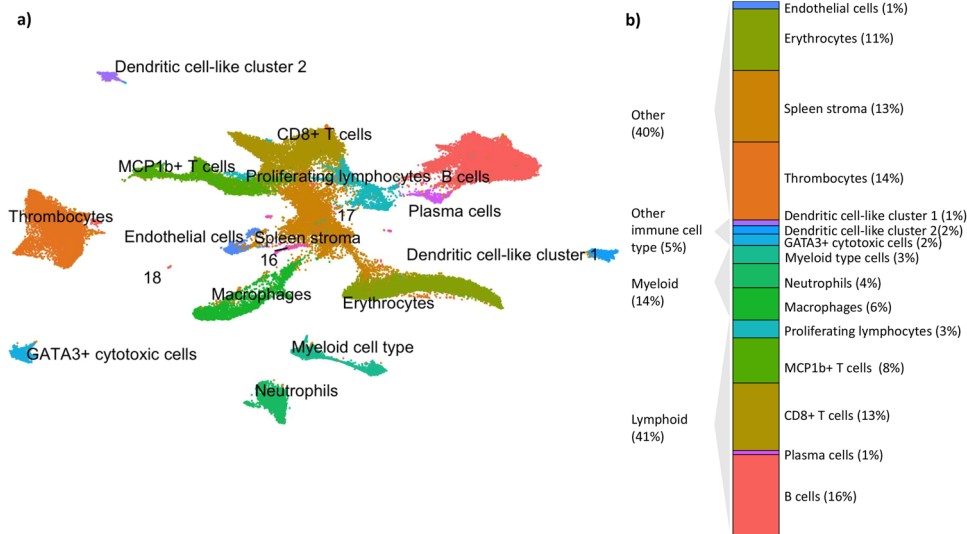

**Fig. 1 Unsupervised clustering of 56,994 Atlantic cod splenic cells. a** Single-cell RNA sequencing of splenic cells is visualised in two dimensions using UMAP. Cluster identity was assigned based upon differentially expressed genes and naming from our last publication[13]. Clusters 16–18 each represent less than 0.2% of the total cells and so have not been named. **b** Bar chart showing percentage of splenic cells as lymphoid cells, myeloid cells, other immune cells, and other splenic cells. Cells from clusters 16–18 have not been included in the percentage calculations.

different from one another, revealing a mixed gradient pattern of gene expression, such as the transcription factors *spi1* and the adaptor protein *sh2d1a* (Fig. 2b). Sub-clusters 2 and 5 expressed the TNF receptor super-family gene *tnfrsf13b*. The expression of shared immunoglobulin genes dominated the most significantly expressed genes for these clusters, with few unique marker genes (Supplementary data 5). B cell sub-cluster 4 differentially expressed *jun-D*, a sub-unit of the AP-1 transcription factor, which has been linked to different roles in B cell activation and differentiation[14].

B cell sub-clusters 5 and 7 had very distinct gene expression profiles. Sub-cluster 5, making up 5% of the B cells, are the previously identified plasma cells and expressed the marker genes *interferon regulatory factor 4* (*irf4*) and *peptidylprolyl isomerase b* (*ppib*) (Fig. 2b) and demonstrated the highest average expression of immunoglobulin genes (Supplementary Fig. 3), as expected. B cell sub-cluster 7, making up only 1% of the B cells, highly differentially expressed several genes, including *thrombospondin 1* (*thbs1*), *thymosin beta 11* (*tmsb11*), and *cellular communication network factor 2* (*ccn2*) (other genes can be seen in Supplementary data 5).

The trajectory analysis Slingshot is an inference tool which computationally orders cell clusters along a predicted developmental trajectory based on gene expression, an ordering known as pseudotime. Performing a Slingshot analysis on the B lymphocyte clusters suggested two lineages; both lineages started in B cell sub-cluster 6, with lineage 1 ending in sub-cluster 5 (the plasma cells) and lineage 2 ending in sub-cluster 7 (Fig. 3a–d). We did not manually select any start or endpoints for the lineages, allowing Slingshot to compute pseudotime in an unbiased fashion. The terminal clusters of the two lineages are consistent with the discrete populations demonstrated by the differential gene analysis. B cell sub-cluster 3, which is the cluster of B cells ordered in pseudotime right before the terminally differentiated plasma cells, highly expressed *proliferation associated 2g4* (*pa2g4*), a gene involved in cell proliferation, differentiation, and survival[15].

We fit a generalised additive model (GAM) to the two predicted trajectories to explore potential differences in the progression of the B cell lineages and identified the top 100 temporally expressed genes (Supplementary data 3). The expression of these genes changed in a continuous manner over pseudotime. The transcription factors *pax5* and *e2a* were present among the top differentially expressed genes identified by GAM and visualised as smooth expression plots (Fig. 3e). The gene *tnfrsf13b* is a gene of biological interest and has also been shown. All three genes are linked to B cell development: *pax5* encodes the B cell lineage-specific activator protein expressed at early stages of B cell differentiation[16], *e2a* plays a role in B and T lymphocyte development[17], and *tnfrsf13b* encodes the transmembrane activator and CAML interactor (TACI) suggested to control B cell differentiation and memory[18,19]. At the end of lineage 1 the expression levels of *pax5* dropped in the plasma cells, while expression levels of *e2a* and *tnfrsf13b* both showed an increase. In lineage 2 the expression levels of the *pax5* and *e2a* remained steady, while expression of *tnfrsf13b* peaked in sub-cluster 2 and then had low expression through the rest of the lineage. It is unclear why there was a noticeable dip in expression level of all *pax5* and *e2a* genes in sub-cluster 1, especially in lineage 1.

**T lymphocyte sub-clusters**. Selecting the 13,753 T lymphocyte cells from the global UMAP and re-clustering them revealed five T cell sub-clusters with distinct gene expression profiles in each (Fig. 4a–c). Expression of *T cell genes* (*tcr*) genes, *cd3d*, and the cytotoxic *granzyme B* (*gzmb*, LOC115532977) genes within the 13,753 cells confirmed all these clusters are T lymphocytes (Fig. 4b). All T cell clusters, except sub-cluster 2, expressed similar levels of *tcra* and *tcrb*. Sub-cluster 2 cells expressed *tcrb* but did not express *tcra*, but instead expressed putative *tcrg* and putative *tcrd*. This cluster also expressed a homologue of *m130*, a scavenger receptor also known as CD163 that is typically considered a marker of activated or anti-inflammatory macrophages[20]. T cell sub-cluster 0 was the only CD8-positive cluster, and also expressed CC chemokine receptor 7 (*ccr7*) and a second granzyme B paralogue (*gzmb*, LOC115532976). Sub-clusters 1–4 were all CD8-negative. Sub-clusters 1 and 4 expressed a homologue of *mcp1b* in especially high levels in comparison to the other sub-clusters. Sub-cluster 1 expressed the chemokine

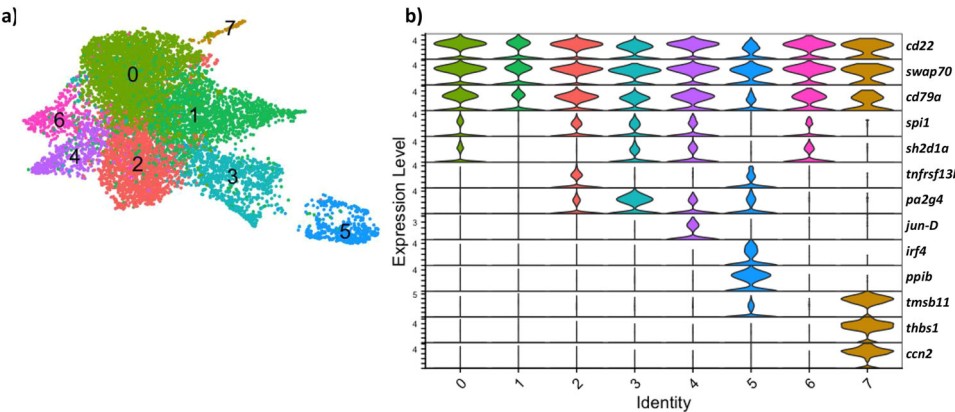

**Fig. 2 Closer examination of 9474 Atlantic cod B lymphocytes reveals 8 sub-clusters with distinct transcriptome profiles. a** Visualisation of B lymphocyte cells using UMAP. **b** Violin plots showing expression of selected genes identified by differentially expressed gene analysis. The y-axis indicates normalised and log-transformed average expression of the selected genes and the x-axis indicates the B cell sub-cluster.

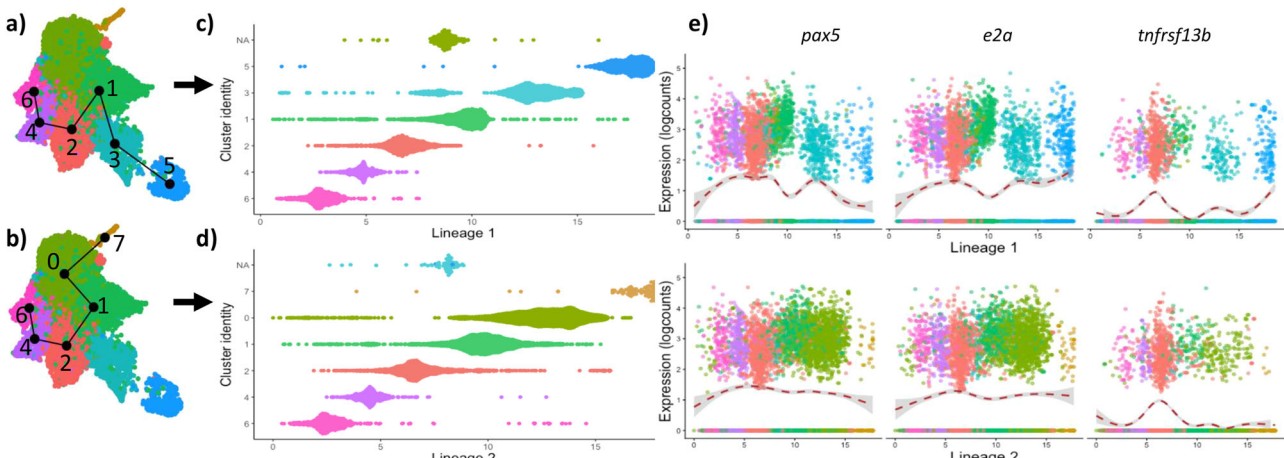

**Fig. 3 Pseudotime analysis of B lymphocyte cells reveals two lineages with different expression patterns of transcription factors and genes. a, b** B cell sub-clusters depicting the Slingshot-predicted lineage trajectories, lineage 1 (**a**) terminating in the B cell sub-cluster 5 (the plasma cells), and lineage 2 (**b**) terminating in B cell sub-cluster 7. **c, d** Depiction of cells within each B cell sub-cluster from the Slingshot-predicted lineage on the y-axis along pseudo-temporal ordering on the x-axis. Each dot represents a cell and its predicted temporal position in development in lineage 1 (**c**) and lineage 2 (**d**). (**e**) Smoothed expression plots along B lymphocyte lineages for the transcription factors *pax5*, *e2a*, and *tnfrsf13b*. Each point represents the expression level of each transcript within a single cell. The colour is consistent with the sub-cluster identity. The red dashed line is a weighted regression to fit a smooth curve through the points (LOESS curve), and the grey area shows a 95% confidence interval.

*ck5b* (*ccl4*) and granzyme k, *gzmk*, while cluster 4 expressed interleukin 18 receptor accessory protein, *il18rap*. T cell sub-cluster 3 expressed the proliferation markers *mki67* and *pcna*, as well as *incenp*, a regulator of mitosis. The top differentially expressed genes of each T cell sub-cluster can be seen in Supplementary data 6. In summary, although five T cell sub-clusters were revealed by the UMAP, it was noticeable that there were three distinct T cell populations: a population that expressed *cd8* (sub-cluster 0), a population that expressed *mcp1b* (sub-clusters 1 and 4), and a population which expressed *m130* (sub-cluster 2) (Fig. 4b–d). Sub-cluster 3 represents proliferating T cells, but their relationship to the other three T cell populations remains inconclusive.

The expression of *cd8*, *m130*, and *mcp1b* does not change with immunisation or infection status over the immune challenge experiment (Supplementary Fig. 4), suggesting these T cells groups are present in both healthy and sick cod. Given the knowledge at hand, we re-analysed the T lymphocytes in our first

scRNA-seq dataset of cod immune cells[13] and searched for the same genes shown in Fig. 4b. We again found that *cd8*, *m130*, and *mcp1b* were expressed in three mutually exclusive populations (Supplementary Fig. 5), echoing the results shown here. This pattern was also seen in the global UMAP containing all the splenic cells (Supplementary Fig. 6).

The only other cells in the spleen to express *mcp1b* or *m130* were the GATA3$^+$ cytotoxic cells (Supplementary Fig. 6). GATA3$^+$ is a master regulator of T helper 2 cells in mammals[21,22] and has also been shown to play a role in the development of innate lymphoid cells[23]. This cluster was identified and named in our previous paper[13], and, consistent with our previous description, did not express other T cell markers, supporting our previous hypothesis that this cluster represents an innate lymphoid cell type.

We carried out a pseudotime trajectory analysis on the T cells as we did for the B cells and show two alternatives in Supplementary Fig. 7. Supplementary Fig. 7A shows a single

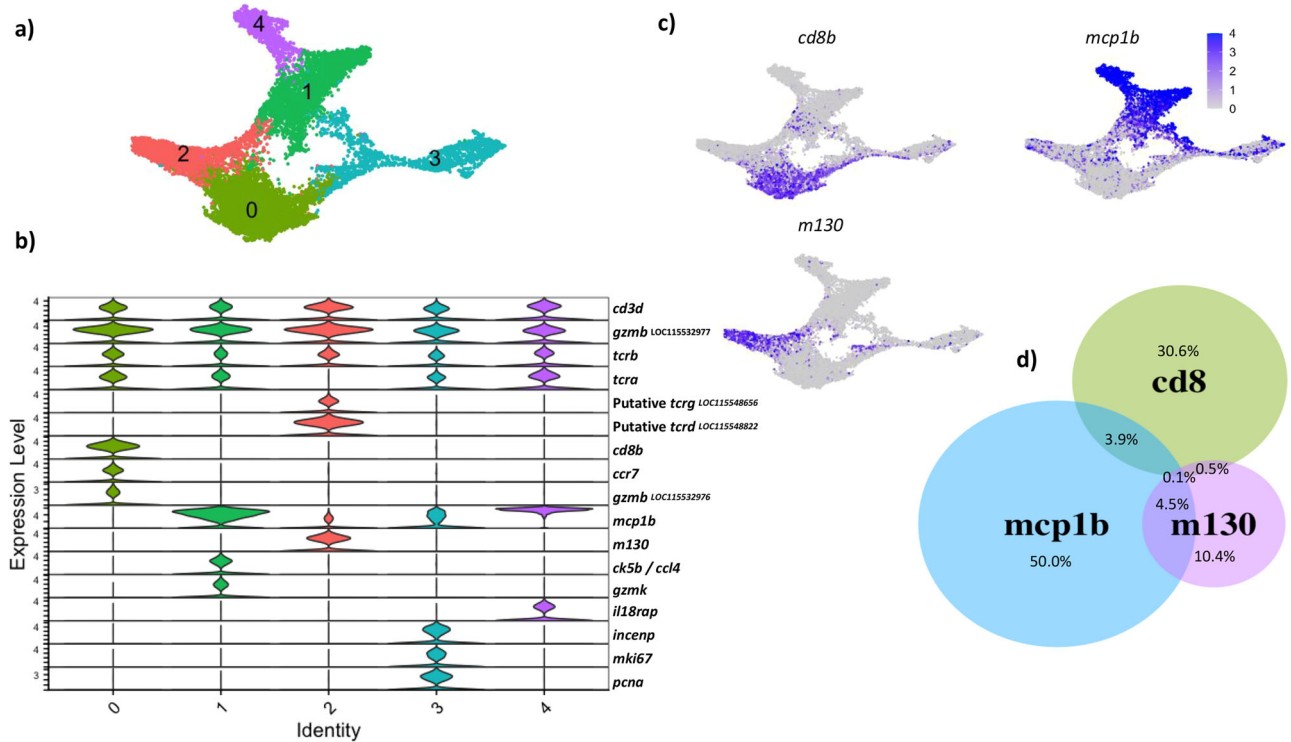

**Fig. 4 Closer examination of 13,753 Atlantic cod T lymphocyte cells reveals 5 sub-clusters with distinct transcriptome profiles.** Expression of T cell co-receptor *cd8b*, cytokine *mcp1b*, and the scavenger receptor *m130* segregates T lymphocytes into three distinct sub-populations. **a** Visualisation of T lymphocyte cells using UMAP reveals five T cell sub-clusters. **b** Violin plots showing expression of selected genes identified by differential expressed gene analysis. The *y*-axis indicates normalised and log-transformed average expression of the selected genes and the *x*-axis indicates the T cell sub-cluster identity. **c** UMAP showing the expression level of *cd8b*, *mcp1b*, and *m130* in T lymphocytes. Cells are coloured by expression level. **d** Venn diagram demonstrating that the expression of *cd8b*, *mcp1b*, and *m130* is largely non-overlapping. The percentage of T cells expressing each gene is shown.

proposed trajectory, starting in sub-cluster 3 (CD8-negative, proliferating cells), then into sub-cluster 0 (CD8-positive), then to sub-cluster 2 (CD8-negative, m130-positive), then in sub-clusters 1 and 4 (CD8-negative, mcp1b-positive). This trajectory seems extremely unlikely. Supplementary Fig. 7B shows the result if we propose multiple trajectories to the algorithm and displays sub-cluster 3 and 4 as separate lineages, while sub-clusters 0, 2 and 1 are linked. This scenario also seems unlikely.

**Phylogenetic overview of *m130*, *mcp1b* and *tcr* genes**. In the phylogenetic tree of *m130* (Supplementary Fig. 8a) we cannot with confidence confirm that the suggested *m130* gene model from the annotation is indeed M130/CD163 as the support within the backbone of the phylogeny was too low. However, there was support to place the gene model within the *group B scavenger receptor cysteine-rich (SRCR) superfamily* and that the gene model was closer to M130/CD163/CD163L than the other superfamily members, making it a M130/CD163-like candidate. Thus, we would describe the gene model *m130* as a scavenger receptor superfamily member that may have M130/CD163-like functions. The classical M130/CD163 property is the ability to bind and internalise low-density lipoproteins[24], such as the haemoglobin-haptoglobin complex[25].

The proposed *mcp1b* gene model clustered together with several other characterised CC chemokines from Atlantic cod (Supplementary Fig. 8b). However, the tree did not resolve beyond clustering the major chemokine family members together, the CC and CXC chemokines respectively, and overall displayed no bootstrap support in the backbone of the phylogeny. This indicated that the *mcp1b*/CCL2 annotation of the gene model is

inaccurate and further characterisation of the gene model is needed. We would thus describe this gene model as a CC chemokine of unknown function.

Phylogenetic analysis of the annotated tcr genes *tcra* (LOC115548821) and *tcrb* (LOC115544273) supported the identity of these genes, while LOC115548656 was grouped with known *tcrg* genes and LOC115548822 was grouped with known *tcrd* genes, suggesting that these two genes represent Atlantic cod *tcrg* and *tcrd*, respectively (Supplementary Fig. 8c)

## Discussion
Multiple genome studies, as exemplified by several teleosts, have revealed a plasticity and adaptability of the vertebrate adaptive immune system. The Atlantic cod[3], and later the entire Gadiformes lineage[4], was shown to have lost the MHC-II/CD4 T cell pathway. Further peculiarities have been found in other teleost species, including the loss of MHC-II/CD4 in pipefish[26,27] and monkfish[28], and the loss of MHC-II/CD4 alone or in combination with the loss of MHC-I/CD8 in several deep-sea species of anglerfish[29]. In the absence of specific antibodies, it is difficult to assess the immune cell composition as a first step towards understanding the immune system adaptations that have occurred in these newly identified adaptive immune systems. Transcriptomic analysis at the single-cell level is a powerful tool that we have used successfully in characterising immune cell populations of the Atlantic cod[13]. In this study, we have vastly increased the number of immune cells studied by scRNAseq, to 57 thousand spleen cells from 34 fish that were sampled during a vaccination and immune challenge study, compared with a total of 7 thousand blood and spleen cells from two fish in our

previous study. The increased number of cells gave us the power to scrutinise in detail the B cell and T cell subsets found within 9474 B cells and 13,753 T cells that were included in the current study.

Unsupervised sub-clustering of B lymphocytes revealed eight B cell sub-clusters, of which the sub-clusters 5 and 7 are transcriptionally most distinct. Sub-cluster 5 is identical with the plasma cell subset that was identified in the global analysis, which included all splenocytes. Trajectory analysis by Slingshot suggested a common maturation/differentiation pathway of B cells encompassing the first four common progenitor B cell sub-clusters, and then diverging into two separate trajectories each ending up in the presumed end-differentiated B cell sub-cluster 5 (the plasma cells) and sub-cluster 7. That one trajectory ends in the plasma cells, a well-established terminally differentiated cell type, gives strength to the unsupervised ordering of the clusters by Slingshot. In addition, B cell sub-cluster 3, the penultimate sub-cluster in the plasma cell lineage, highly expresses *pa2g4*, a gene involved in cell proliferation and differentiation, further strengthening the trajectory analysis. B cell sub-cluster 7, which is the endpoint of the other trajectory makes up only 1% of the B lymphocytes. This cluster expresses *thbs1*, *tmsb11*, and *ccn2* at a high level, which leads us to speculate that this small subset of highly differentiated B cells may have effector functions that involve communication with other cells. This functionally differentiates this cluster from plasma cells, whose function involves antibody production, and so may be more in line with the regulatory functions of certain B cell subsets that have been recently suggested in mammalian immunology[30]. Functional confirmation of this sub-cluster of B cells is needed when appropriate reagents become available.

The most remarkable finding in the T lymphocyte population is the absence of T cell co-receptor *cd8* expression in several subsets, accounting for 65% of all T cells. The majority of the CD8-negative T cells are positive for a CC chemokine initially annotated as *mcp1b*. However, great differences exist between fish and mammalian cytokines, and within fish species, making annotation of cytokines extremely problematic[31–33]. The sequence phylogeny of *mcp1b* gives support that this gene is a C-C motif chemokine but gives little to support this gene model is MCP-1/CCL2. Cytokines rapidly adapt in species and also experience prevalent gene losses and expansions[33]. Lacking functional data and a clear naming alternative we have kept the naming given in gadMor3: *mcp1b*. MCP-1 is a chemoattractant for monocytes and T cells in humans[34]. In humans and mice, the main producers of MCP-1 are myeloid cells (monocytes and macrophages)[35], in addition it is also produced by some non-immune cells such as epithelial and endothelial cells[36]. In our scRNA-seq data of the Atlantic cod spleen, we find that *mcp1b* is primarily expressed by T cells and GATA3$^+$ cytotoxic cells, and we do not find *mcp1b* expression in the myeloid or stromal cells present in the spleen, further suggesting that *mcp1b* as expressed in Atlantic cod is different from the role of MCP-1 in mammals and does not fulfil the same role. We do not know which genes encode the receptor(s) for *mcp1b* in cod, or which cells that express them. It is intriguing to note that expression of *gzmb* is slightly lower in the same clusters that show high expression levels of *mcp1b*, sub-clusters 1, 3, and 4. It is thus tempting to speculate that cells in these sub-clusters have reduced cytotoxic functions and rather may be involved more heavily to signalling with other not-yet identified immune cells through its secretion of the chemokine *mcp1b*.

Most of the T cells which are negative for both CD8 and MCP1b express *m130*. Phylogenetic comparisons of this gene demonstrated that *m130* clustered within the group B SRCR superfamily, and we propose it is CD163-like. In humans and

rodents, CD163 is found to be expressed exclusively in cells of monocytic origin[37], which is not found here. The B SRCR superfamily is an ancient and conserved protein domain, and some orthologues genes within this family, CD163c-α-like/WC1, have been shown to play a role in the regulation of gamma delta T cells in cattle[38].

In our scRNA-seq data from cod spleen, *cd8*, *mcp1b*, and *m130* are expressed in T lymphocytes in a largely exclusive manner, giving rise to three distinct and separate T cell sub-populations. The CD8-positive, MCP1b-positive, and M130-positive populations all express key T cell markers, such as *cd3*, *tcr* genes, and *gzmb* (Fig. 4a). There are two developmental scenarios for these three distinct populations: they have either acquired the expression of their respective markers during lymphogenesis in the thymus and exit thymus as three populations of mature T cells that do not interconvert in the periphery; or that they all evolve from the same mature progenitor T cells and represent three differentiation stages. The finding that M130-positive cells uniquely express two genes putatively suggested to be *tcrg* and *tcrd*, indicates that the M130-positive cells represent a separate lineage of T cells. More detailed molecular analysis such as full-length TCR sequencing is needed to ascertain whether these cells may represent gamma delta T cells, the existence of which has not been demonstrated in cod before. Carrying out a trajectory analysis on the T cells did not resolve into biologically meaningful lineages. Slingshot typically requires supporting evidence with proposed start or endpoints or the number of trajectories in the cells present. Thus, we currently lack the resolution to propose the possible T cell trajectories on our current data. ScRNA-seq combined with single-cell TCR sequencing may also reveal the developmental relationship between the various T cell sub-populations, as well as whether some sub-clusters may contain unconventional T cells similar to natural killer T cells or mucosal associated invariant T cells that show restricted and semi-invariant V-gene usage in mammalian systems[39,40]. The CD8-positive population uniquely expresses chemokine receptor *ccr7* suggesting this subset may have a different trafficking pattern than the CD8-negative T cell subsets.

The B and T cell differentiation pathways in teleosts are poorly understood, and there may be substantial differences between different lineages of teleosts. An alternative to use of classical surface markers is to identify transcription factors which are differentially expressed along predicted lineages. *pax5* and *e2a* are well-known genes participating in B cell differentiation and activation[16,17], and have even been identified in teleost B cell lymphogenesis[41]. *pax5* and *e2a* are among the top differentially expressed genes in the predicted B cell lineages in this study, suggesting these genes also play an important role in B cell differentiation in the Atlantic cod. Expression of the gene *tnfrsf13b* was shown to be increased in the cod plasma cells, results in alignment with human studies suggesting *tnfrsf13b* has a role in the development of plasma cells and production of much of the Ig in blood[18,19]. The TACI surface receptor encoded by this gene is suggested to be involved in T cell-independent antibody responses[42], which is interesting given the absence of CD4 T helper cells in cod.

It is interesting to note that the two novel T cell population markers we have discussed here, namely *m130* and *mcp1b*, are only expressed in the GATA3$^+$ cytotoxic cell population in addition to the T cells. Despite sharing these unique markers, the GATA3$^+$ cytotoxic cells do not express *tcr* and are transcriptionally distant from the other lymphocyte clusters, as visualised by the global UMAP, and so we cannot classify them as T cells. We previously hypothesised these cells represent a population of innate lymphoid cells[13], and the additional findings here make this a population of interest for future studies.

Here we present a refined characterisation of the T and B lymphocytes in the Atlantic cod spleen using single-cell transcriptomics. The discovery of novel T cell subset markers gives guidance to the future direction of studying T cell development, differentiation, and function in Atlantic cod. In addition, the trajectory analysis reveals candidate transcription factors that drive B cell differentiation. These hypotheses need to be validated in future studies.

## Methods

**Atlantic cod sampling**. Atlantic cod sampled for this study originate from a single breeding family (bred from 1 female and 2 males) from the NOFIMA breeding programme in Tromsø, Norway. They were reared, vaccinated, and challenged at the NIVA Research Facility at Solbergstrand, outside Oslo, Norway. The rearing and sampling were performed according to the European animal welfare regulations and approved by the Norwegian authorities (FOTS ID 21758).

Sampling took place between 9 and 11 months of age (average length = 21 cm, range 13–27 cm). The water temperature was maintained at ~8 °C in keeping with the seasonal water temperature, with a water salinity of 34 PSU, light conditions were 12:12 hour light:dark, and the cod were fed with Skretting cod pellets. Spleen samples were taken from 34 Atlantic cod that were used in a vaccination study for *V. anguillarum*. Bath vaccination and boost were carried out using formalin-fixed *V. anguillarum* serotype O2a (ATCC19264 strain), and the immune challenge was carried out with the same strain 8 weeks after boost, with a bacterial load of $1.1 \times 10^7$ CFU/ml. Samples were taken before and after vaccination priming and boosting, before immune challenge, then day one, seven, and 15 after immune challenge for naive and vaccinated fish. Vaccination priming and boost, and immune challenge were carried out by bath immersion. A schematic demonstrating sampling is shown in Supplementary Fig. 1. Fish were killed by cranial concussion and tissue sampling was conducted within minutes of fish death. As blood was also taken for additional investigations not mentioned in this paper, gill slit could not be carried out. The spleens were removed and placed in PBS-BSA 0.01% solution and kept on ice. Spleen cell suspensions were obtained by gently forcing the tissue through a cell strainer (Falcon, 100 μm) and diluted to 200 cells/μl in fresh PBS-BSA 0.01% solution. All cells were kept in regular microcentrifuge tubes to minimise cell loss and kept on ice for transport and laboratory work.

**Single-cell cDNA library preparation and sequencing**. Single-cell RNA sequencing (scRNAseq) and the preparation of the libraries were performed according to published protocols[13,43,44]. A droplet sequencing generator (Dolomite) individually encapsulated cells with barcoded beads. Libraries were sequenced at the Norwegian Sequencing Centre (Oslo University Hospital), on the NextSeq500 platform with a 75 bp kit, high output mode, with paired-end reads. 20 bp were sequenced in Read 1 using a custom sequencing primer (GCCTGTCCGCGGAAGCAGTGGTATCAACGCAGAGTAC) and 60 bp in Read 2 with the regular Illumina sequencing primer. The sequencing data is available at the ENA repository with Accession number PRJEB47815.

We used the Drop-seq Core Computational Protocol[45] to demultiplex the raw sequencing reads and map them to the most recent version of the Atlantic cod genome, gadMor3 (RefSeq accession GCF_902167405.1) using STAR alignment. A gene of interest, *gata3*, was present in gadMor2[46] but missing in gadMor3, so the *gata3* gene sequence was manually added to the gadMor3 assembly fasta file. The reads were summarised into gene expression matrices using the Drop-seq programme 'DigitalExpression'.

**Pre-processing workflow**. The resulting count matrices for each sample were loaded into Seurat (version 4.0.2)[47–50]. The count matrices from each of the 34 cod were integrated into one Seurat object through anchor identification using the 'FindIntergrationAnchors' and 'IntergrateData' Seurat functions. The percentage of mitochondrial gene expression was added by 'PercentageFeatureSet' and cell barcodes with more than 5% of mitochondrial genes, 'percent.mt' were removed. Cell barcodes with fewer than 150 genes and greater than 1500 genes, 'nFeature_RNA', and those that contained genes expressed in less than three cells, 'min.cells', were filtered out. Additionally, cells with a total number of biological molecules greater than 4000, 'nCount_RNA', were removed. These steps reduce the number of low-quality cells and reduce cell multiplets or cell barcodes which do not contain a true cell transcriptome but rather ambient RNA. Following quality control filtering, we derived a gene expression matrix of 19,279 genes across 56,994 splenic cells. An overview of the samples in each treatment group in the immune challenge protocol, sequencing library, average mapping percentages, number of cells, mapped transcripts, and genes are shown in Supplementary data 1.

After filtering, data were normalised using the 'NormalizeData' function, variable features were identified using the 'FindVariableFeatures' function, and features were scaled using the 'ScaleData' function. PCA analysis was performed, and the most significant principal components were identified by plotting the first 50 dimensions on an Elbow plot (Supplementary Fig. 2).

**Dimension reduction and assigning cluster identity**. Unsupervised dimension reduction was performed using Uniform Manifold Approximation and Projection (UMAP)[51] on the first 40 principal components and the clusters were identified using the 'FindNeighbors' (dims = 1:40) and 'FindClusters' functions (res = 0.2). UMAP works by placing cells on a two-dimensional map based on their expression profile. Similar cells are placed together, while dissimilar cells are placed further apart. Thus, cells with similar transcriptional profiles form a 'cluster' and can be assumed to be a particular cell type. The distances between clusters also reflect similarity between clusters, meaning similar cell types, or clusters, are closer together. The number of different clusters displayed is dependent on parameter settings (i.e., the number of dimensions and the resolution). We decided on settings where each cluster has well-defined marker genes, more or less exclusively expressed by this cluster only. Cluster identities were assigned by assessing differential expression of marker genes between different clusters using the bio-markers detected from the 'FindAllMarkers' function (only.pos = TRUE, min.pct = 0.25, logfc.threshold = 0.25). The top 20 differentially expressed genes (DEGs) for all clusters found in the splenic samples can be seen in Supplementary data 2. Each cluster was named by examining these DEGs and referring to gene markers found in the literature.

To examine the lymphocyte populations more closely, we selected the B cells, CD8-positive T cells, MCP1b-positive T cells, proliferating lymphocytes, and plasma cells from the global splenic UMAP, thus selecting 23,227 lymphocytic cells. These lymphocytes were then clustered on the first 40 dimensions at 0.2 resolution. As a result of this unsupervised clustering step, the proliferating lymphocytes cluster was divided into B lymphocytes or T lymphocytes. The 9474 cells in total identified as B lymphocytes were clustered using UMAP on the first 40 dimensions with a 0.3 resolution. The 13,753 cells identified as T lymphocytes were clustered using UMAP on the first 40 dimensions with a 0.09 resolution.

**Trajectory analysis**. We used Slingshot (v. 1.8.0) analysis[52] to infer possible differentiating trajectories within the lymphocyte sub-clusters. UMAP was used to determine the dimensionality and the lineages were constructed in an unsupervised way. To infer differential gene expression in the lineages predicted by the Slingshot analysis, we used the GAM package (v. 1.20) to run a generalised additive model (GAM) which uses a LOESS term for pseudotime. This identified the top 100 most variable genes across pseudotime (Supplementary data 3). The lineages and expression of key genes were visualised using the Slingshot tools, the ggplot2 (v. 3.3.3)[53] package, and the scater package (v. 1.18.6)[54].

**Phylogeny analysis of *m130, mcp1b,* and *tcr* genes**. All gene names reported in the scRNA analyses are derived from the automated annotation of the gadMor3 assembly. To further explore the identity of particular genes of interest, *mcp1b* (LOC115529242), *m130* (LOC115541469), and the *tcr* (T-cell receptor) genes, in more detail, maximum likelihood phylogenetic trees were constructed.

Scavenger Receptor Cysteine-Rich Type 1 Protein M130 (M130/CD163) references were collected from the groups Mammalia, Aves, Reptilia, Amphibia, and Teleostei at Uniprot.org where available. The predicted gene models of the scRNA-reported annotations were then subjected to a blastp at NCBI towards mammalian and teleostei databases using default parameters. Here, 3–5 hits annotated as M130/M160/CD163 were selected to be included in the multiple sequence alignment. The scavenger receptor cysteine rich (SRCR) domain present in M130/CD163 is an ancient and conserved domain shared by members of the group B SRCR superfamily[38]. To provide more framework to the alignment of *m130*, additional members of the group B SRCR family were included (CD5, CD5L and CD6) together with two other scavenger receptors: scavenger receptor cysteine-rich family member with 4 domains (SSC4D) and Scavenger Receptor Class B Member 1 (SCARB1), all obtained in the same manner as described above.

Monocyte chemotactic protein 1 (MCP-1/CCL2) references were obtained as described above for CD163. Furthermore, as MCP-1/CCL2 is a chemokine belonging to a large gene family, a set of references for the most common CC chemokines in humans and fish were downloaded from GenBank and added to the alignment. In addition, some CXC chemokines were added to function as a collective outgroup. Finally, complete CC chemokine characterisations from Channel catfish and Atlantic cod were added, together with all predicted gene models from gadMor3 reported from a tblastn search using the MCP-1/CCL2 references, default parameters, and e-value cutoff at 1e-1[55].

The final multiple sequence alignments (protein) were aligned using MUSCLE in MEGA7 and default parameters[56]. The resulting protein alignment was subjected to a maximum likelihood run using raxml-ng[57] after model testing using modeltest-ng[58]. For M130/CD163, the best scoring model was WAG + G4 and for MCP-1/CCL2 VT + I + G4. Both maximum likelihood runs were run until convergence and followed with 500 bootstrap replicates.

We also constructed a phylogenetic tree of *tcr* genes in Atlantic cod and in other selected species. The annotated *tcrb* (LOC115544273) and *tcra* (LOC115548821) genes in gadMor3, as well as putative *tcr* genes (LOC115548656 and LOC115548822) identified by blasting gadMor3 with *tcr* genes from different teleosts, reptiles, and birds, were compared. All genes with hits above 1e-100 were collected and added to the overall sequence alignment. A multiple sequence alignment was generated using MEGA7 and its MUSCLE alignment programme with default setting. From here a neighbour-joining tree using Poisson distribution,

pairwise deletion and 200 bootstrap replicates was made again using MEGA7. The trees are visualised and edited for presentation purposes using FigTree v 1.4.4[59].

**Reporting summary**. Further information on research design is available in the Nature Research Reporting Summary linked to this article.

## Data availability

The sequencing data is available at the ENA repository with Accession number PRJEB47815. The Rscript used to analyse the data is publicly available at https://github.com/NaomiCroft/Rscript-analysing-Atlantic-cod-lymphocytes.

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

## Acknowledgements

Financing from the University of Oslo Life Sciences Convergence programme to the project "Comparative immunology of fish and humans (COMPARE)" in collaboration with the "rational design of vaccination strategies for Atlantic cod (VACSACOD)" programme, Research council of Norway. The Norwegian Sequencing Centre (NSC: https://www.sequencing.uio.no) provided sequencing. We thank Henning Sørum (Norwegian University of Life Sciences) for discussion and planning of the immunisation experiments

## Author contributions

N.C.G.—First author and corresponding author. Responsible for laboratory work, data analysis, producing all figures and most of the supplementary figures, and responsible for the writing of the paper. A.K.K.—Gave advice on data analysis and bioinformatic support. Read the paper and suggested improvements. S.F.N.—Involved in planning of experimental set-up and sample collection from Atlantic cod. Read the paper and suggested improvements. M.H.S.—Carried out phylogenetic analysis on key genes of interest: mcp1b, m130, tcrd and tcrg. Made supplementary figures 8a–c. Read the paper and suggested improvements. K.S.J.—Involved in planning stages of the experimental set-up. Read the paper and suggested improvements. F.E.J.—Involved in planning of the experimental set-up and sample collection from Atlantic cod. Gave advice on data analysis and structure of the paper. Read the paper and suggested improvements. S.W.Q.—Last author and corresponding author. Involved in planning the experimental set-up, sample collection, and laboratory procedures. Gave advice on data analysis and structure of the paper. Read the paper and suggested improvements.

## Competing interests

The authors declare no competing interests.
