## [Peer Review File · Communications Biology]

Reviewers' comments:

Reviewer #1 (Remarks to the Author):

In their manuscript, Guslund et al. aimed to unravel the heterogeneity of lymphocyte subsets in the spleen of Atlantic cod, using single-cell RNA sequencing. Based on the data obtained from 34 fish at 12 timepoints during a vaccination and immune challenge, the authors analyzed the expression of 56 994 splenic cells. They identified five TCR-positive T cell subsets and eight BCR-positive B cell subsets. Furthermore, using the well-designed vaccination and challenge experiment, the authors further evaluated the trajectory of B cell differentiation during the immune responses. The data presented in the current manuscript represent a massive leap in our current understanding of teleost immunology and provide valuable insights into the basic mechanisms of immune response in the absence of MHC II and CD4-positive T cells, which have been lost in Gadiformes approximately 100 million years ago.

The manuscript is well designed, the data were analyzed appropriately and visualized accordingly. Overall, the manuscript is well written and presents valuable information. I do have only a couple of minor remarks for the authors to consider:

A) In section 3.2. authors focus on the heterogeneity of the B lymphocytes and identified eight sub-clusters with a shared expression of the marker genes *cd22*, *swap70*, *cd79a*, and immunoglobulin genes. I wonder whether authors could further elaborate on the proportion and identity of the B cells according to the profile of their immunoglobulin expression. From the described results, it is not clear whether the B cells expressed the IgM solely or whether the IgT/Z and IgD were also found in the transcripts.

B) Section 3.3 focusing on the phylogenetic overview of *m130*, *mcp1b*, and *trc* seems out of context and is justified only by the following section focusing on the clustering of T lymphocytes. Hence, for better clarity, I would recommend combining both sections, first introducing the T cell sub-clusters and only then elaborating on the identity of *mcp1b* and *m130*. Furthermore, since authors themselves express certain doubts about the annotation of these genes, it would be helpful to describe these T cell markers as *mcp1b* and *m130*, rather than mentioning their mammalian equivalents CD163 and CCL2 in text and the figures.

C) Since authors elaborate extensively on the trajectory analysis of B cell sub-clusters, I am somewhat surprised they did not attempt to perform the trajectory analysis for the identified T cell sub-clusters to gain deeper insights into the functional relevance of these subsets for the development of immune response. Taking advantage of the expression signatures of 13 753 T cells, wouldn't it be possible to dissect their key expression profiles further and identify the driving transcription factors, as shown in the example of B cells.

Reviewer #2 (Remarks to the Author):

Dear Editor,

The manuscript entitled "Lymphocyte subsets in Atlantic cod (*Gadus morhua*) interrogated by single-cell sequencing" by Guslund et al. presents the study of lymphocyte heterogeneity in Atlantic cod spleen using single-cell RNA sequencing. The authors describe TCR-positive T cell subsets and BCR-positive B cell subsets and propose a B cell trajectory of differentiation. They also identified a T cells subpopulation that are CD8-negative and noticed that most of those CD8-negative T lymphocytes express the homolog of monocyte chemotactic protein 1b, while another subset of CD8-negative T lymphocytes expresses the homolog of the scavenger receptor *m130*. In my opinion, the manuscripts' objective and perspective are very interesting, the manuscript is well-written, the experiments are well designed and the results have been adequately analyzed. My detailed comments for the authors to consider are provided below:

1. In page 2, line 55 the authors state that the before mentioned model is "in general not validated in teleosts". It would be useful to add whether the model is validated in specific cases, i.e. is it validated in another fish species? Or is part of it validated in a study?

2. In page 5, lines 213-215, a figure or a table would be beneficial to support that claim.
 3. In page 5, line 218, supplementary excel sheet 4 contains B cell subclusters DEGs. Please provide the correct reference excel sheet (i.e. excel sheet 6?).
 4. In page 5, lines 219-220, why the authors mention CD8-negative T cells? The figure contains CD8+ T cells
 5. In page 6, line 259, supplementary excel sheet 5 contains T cell subclusters top DEGs. Please provide the correct reference excel sheet.
 6. In page 8, line 313, I would like to read some rational for choosing these specific genes for phylogenetic analysis in that point of the manuscript. I understand why the authors chose them by reading the whole manuscript but I believe that the addition of one or two sentences about it here would be beneficial for the readers.
 7. In page 8, line 335, and throughout the text five T cell sub-clusters are mentioned. However, in figure 4 7 sub-clusters are clearly presented. Please explain.
 8. In page 8, line 340, the m130 is not presented in figure 4B but in figure 4A. Is there a mistake or I didn't understand the authors' comment? In any case, a clarification is needed.
 9. In page 8, line 342, shouldn't ccr7 be included in figure 4?
 10. In page 8, line 344, based on figure 4B ccl4 seems to be expressed in sub-clusters 2 and 3, not 1. Also, gzmk is expressed in sub-cluster 3 not 1. Analogously, mki67 and pcna are not expressed in fraction3. Please correct.
 11. In page 8, line 347, supplementary excel sheet 6 doesn't contain T cell subclusters DEGs. Please provide the correct reference excel sheet.
 12. In page 9, lines 350-351, mcp1b and m130 are attributed to different sub-clusters from those shown in figure 4B and 4C.
 13. In figure 4, please clarify whether CD163 refers to m130 and CCL2 to mcp1b or something else. The nomenclature in figure and legend/ text are different and very confusing.
 14. In page 9, lines 371-376, the GATA3+ cells are not obvious to me, neither explained. Please clarify.
 15. The suppl. figure 7 is not mentioned anywhere.
 16. In page 10, lines 412-413, the absence of T cell cd8 expression in several subsets is not well described in the results section, in my opinion. Please revise it in order to be easier for the reader to follow the discussion about it.
- Overall, it is a very well written and informative manuscript, in my opinion.

Reviewer #3 (Remarks to the Author):

The paper was overall well written and a very interesting topic for comparative immunologists, using cutting edge molecular methodologies. For those not familiar with snSeq there is no specific description of how the cell types are broken down. What determines their grouping. How did the authors decide

what was a macrophage versus a dendritic cell, or any of the cell types how were they initially separated. I understand this was done in a previous publication, but some brief explanation would help in this study. What about the presence of memory B and T cells? Are there markers or expression profiles to suggest this and/or the lack of them? The presence of these cell populations should be discussed.

1-How was the 'vaccine' prepared? And was the 'vaccine' the same ATCC strain as the challenge bacteria?

2-Could the pseudotime analysis be conducted for T-cell differentiation as well – if not explain? And does the 'pseudotime clock' for differentiation of B-cells potentially agree with the expression profiles over time following antigen stimulation? I understand there are limited samples looked at during this snSeq, but either the authors could look at an expanded sample set or at least examine the data they have over time from this study to suggest whether the pseudotime analysis for differentiation of B-cell populations is in agreement with what would be expected in the spleen following antigen challenge over time?

Line 413 – ‘subsets’

Line 416 – ‘...is extremely problematic’

Reviewers' comments:

Reviewer #1 (Remarks to the Author):

In their manuscript, Gusslund et al. aimed to unravel the heterogeneity of lymphocyte subsets in the spleen of Atlantic cod, using single-cell RNA sequencing. Based on the data obtained from 34 fish at 12 timepoints during a vaccination and immune challenge, the authors analyzed the expression of 56 994 splenic cells. They identified five TCR-positive T cell subsets and eight BCR-positive B cell subsets. Furthermore, using the well-designed vaccination and challenge experiment, the authors further evaluated the trajectory of B cell differentiation during the immune responses. The data presented in the current manuscript represent a massive leap in our current understanding of teleost immunology and provide valuable insights into the basic mechanisms of immune response in the absence of MHC II and CD4-positive T cells, which have been lost in Gadiformes approximately 100 million years ago.

The manuscript is well designed, the data were analyzed appropriately and visualized accordingly. Overall, the manuscript is well written and presents valuable information. I do have only a couple of minor remarks for the authors to consider:

Question 1) In section 3.2. authors focus on the heterogeneity of the B lymphocytes and identified eight sub-clusters with a shared expression of the marker genes *cd22*, *swap70*, *cd79a*, and immunoglobulin genes. I wonder whether authors could further elaborate on the proportion and identity of the B cells according to the profile of their immunoglobulin expression. From the described results, it is not clear whether the B cells expressed the IgM solely or whether the IgT/Z and IgD were also found in the transcripts.

Answer 1) The Atlantic cod does not have IgT/Z based on genomic data. In our data, there are roughly 20 Ig genes that are highly expressed in the B cells, of these 3 are IgH genes. Of these, 2 are IgM and one is annotated as IgE. IgE is not an immunoglobulin found in teleosts, so this is probably an annotation error. In conclusion, we have IgM but have not found IgD, although we cannot rule this out. We hope to look into the Ig sequences more deeply in future studies and do not delve into this in our current manuscript.

Question 2) Section 3.3 focusing on the phylogenetic overview of *m130*, *mcp1b*, and *trc* seems out of context and is justified only by the following section focusing on the clustering of T lymphocytes. Hence, for better clarity, I would recommend combining both sections, first introducing the T cell sub-clusters and only then elaborating on the identity of *mcp1b* and *m130*. Furthermore, since authors themselves express certain doubts about the annotation of these genes, it would be helpful to describe these T cell markers as *mcp1b* and *m130*, rather than mentioning their mammalian equivalents CD163 and CCL2 in text and the figures.

Answer 2) This is a sensible suggestion, thank you. We have moved the section 'Phylogenetic overview of *m130*, *mcp1b* and *trc* genes' to read underneath the 'T lymphocyte sub-clusters' section. We have also updated the figures to be consistent with naming of *m130* and *mcp1b*. Now starting line 477.

Question 3) Since authors elaborate extensively on the trajectory analysis of B cell sub-clusters, I am somewhat surprised they did not attempt to perform the trajectory analysis for the identified T cell sub-clusters to gain deeper insights into the functional relevance of these subsets for the development of immune response. Taking advantage of the expression signatures of 13 753 T cells, wouldn't it be possible to dissect their key expression profiles further and identify the driving transcription factors, as shown in the example of B cells.

Answer 3) Again, this is a very sensible suggestion and one that we had originally undertaken. We

had removed this analysis from our manuscript as the resultant trajectories seemed speculative. We have added the original analysis back in as supplementary figure 8, with a short description added in the results section starting line 469, and in the discussion starting line 578.

Reviewer #2 (Remarks to the Author):

Dear Editor,

The manuscript entitled “Lymphocyte subsets in Atlantic cod (*Gadus morhua*) interrogated by single-cell sequencing” by Guslund et al. presents the study of lymphocyte heterogeneity in Atlantic cod spleen using single-cell RNA sequencing. The authors describe TCR-positive T cell subsets and BCR-positive B cell subsets and propose a B cell trajectory of differentiation. They also identified a T cells subpopulation that are CD8-negative and noticed that most of those CD8-negative T lymphocytes express the homolog of monocyte chemotactic protein 1b, while another subset of CD8-negative T lymphocytes expresses the homolog of the scavenger receptor m130.

In my opinion, the manuscripts’ objective and perspective are very interesting, the manuscript is well-written, the experiments are well designed and the results have been adequately analyzed. My detailed comments for the authors to consider are provided below:

Question 1. In page 2, line 55 the authors state that the before mentioned model is “in general not validated in teleosts”. It would be useful to add whether the model is validated in specific cases, i.e. is it validated in another fish species? Or is part of it validated in a study?

Answer 1) We have added a couple of sentences describing what is known for teleost affinity selection to better explain this statement (lines 56-62).

Question 2. In page 5, lines 213-215, a figure or a table would be beneficial to support that claim.

Answer 2) Apologies that this was not clear. This is provided in *Supplementary excel sheet 4* (referenced line 263).

Question 3. In page 5, line 218, supplementary excel sheet 4 contains B cell subclusters DEGs. Please provide the correct reference excel sheet (i.e. excel sheet 6?).

Answer 3) Yes, thank you. There has been a mix up with some of the supplementary excel sheets. These have been corrected.

Question 4) In page 5, lines 219-220, why the authors mention CD8-negative T cells? The figure contains CD8+ T cells

Answer 4) The figure shows both CD8+ T cells and MCP1b+ T cells (The CD8-negative T cells). We agree this is confusing and have reworded (line 65).

Question 5. In page 6, line 259, supplementary excel sheet 5 contains T cell subclusters top DEGs. Please provide the correct reference excel sheet.

Answer 5) Same answer as question 3.

Question 6. In page 8, line 313, I would like to read some rational for choosing these specific genes for phylogenetic analysis in that point of the manuscript. I understand why the authors chose them by reading the whole manuscript but I believe that the addition of one or two sentences about it here would be beneficial for the readers.

Answer 6) This agrees with a comment from reviewer 1, and we have chosen to move this section below the ‘T lymphocytes sub-clusters’ section to give a better flow of information. Now starting line 477.

Question 7. In page 8, line 335, and throughout the text five T cell sub-clusters are mentioned. However, in figure 4 7 sub-clusters are clearly presented. Please explain.

Answer 7) Apologies for confusion, the submitted Figure 4 was an old figure. This has been updated.

Question 8. In page 8, line 340, the m130 is not presented in figure 4B but in figure 4A. Is there a mistake or I didn't understand the authors' comment? In any case, a clarification is needed.

Answer 8) Thank you, we have updated this wording to Figure 4A.

Question 9. In page 8, line 342, shouldn't ccr7 be included in figure 4?

Answer 8) Same answer as for question 7.

Question 10. In page 8, line 344, based on figure 4B ccl4 seems to be expressed in sub-clusters 2 and 3, not 1.

Answer 10) Same answer as for question 7.

Question 11. Also, gzmk is expressed in sub-cluster 3 not 1. Analogously, mki67 and pcna are not expressed in fraction3. Please correct.

Answer 11) Same answer as for question 7.

Question 12. In page 8, line 347, supplementary excel sheet 6 doesn't contain T cell subclusters DEGs. Please provide the correct reference excel sheet.

Answer 12) Same answer as question 3.

Question 13. In page 9, lines 350-351, mcp1b and m130 are attributed to different sub-clusters from those shown in figure 4B and 4C.

Answer 13) Same answer as for question 7.

Question 14. In figure 4, please clarify whether CD163 refers to m130 and CCL2 to mcp1b or something else. The nomenclature in figure and legend/ text are different and very confusing.

Answer 14) Same answer as for question 7.

Question 15. In page 9, lines 371-376, the GATA3+ cells are not obvious to me, neither explained. Please clarify.

Answer 15) We agree supplementary Figure 6 is not clear and have adapted the figure to show the location of the GATA3+ cells.

Question 16. The suppl. figure 7 is not mentioned anywhere.

Answer 16) Thank you, this has been added. (Due to rearranging sections this is now supplementary figure 4, line 457).

Question 17. In page 10, lines 412-413, the absence of T cell cd8 expression in several subsets is not well described in the results section, in my opinion. Please revise it in order to be easier for the reader to follow the discussion about it.

Answer 17) Thank you, we have tried to incorporate your suggestion into the results section to improve clarity (Lines 372-374).

Overall, it is a very well written and informative manuscript, in my opinion.

Reviewer #3 (Remarks to the Author):

The paper was overall well written and a very interesting topic for comparative immunologists, using cutting edge molecular methodologies.

Question 1) For those not familiar with snSeq there is no specific description of how the cell types are broken down. What determines their grouping. How did the authors decide what was a macrophage versus a dendritic cell, or any of the cell types how were they initially separated. I understand this was done in a previous publication, but some brief explanation would help in this study.

Answer 1) We have added an explanation of cell clustering in the results section 2.4 (lines 157-184)

Question 2) What about the presence of memory B and T cells? Are there markers or expression profiles to suggest this and/or the lack of them? The presence of these cell populations should be discussed.

Answer 2) We find proliferating B and T cells but do not know if they represent possible memory populations, these in general are not validated in teleosts. We did not find populations that have a DEG list suggesting they are memory populations.

Question 3) How was the 'vaccine' prepared? And was the 'vaccine' the same ATCC strain as the challenge bacteria?

Answer 3) Thank you, we have added in a sentence to the methods, section 2.1, clarifying this (line 99-101).

Question 4) Could the pseudotime analysis be conducted for T-cell differentiation as well – if not explain?

Answer 4) In agreement with comments from reviewer 1, it is a sensible suggestion to include a T cell trajectory analysis and one that we had originally undertaken. We had removed this analysis from our manuscript as the resultant trajectories seemed speculative. We have added the original analysis back in as supplementary data, with a short description added at the end of the 'T lymphocyte sub-clusters section .In results section starting line 469, and in the discussion starting line 578.

Question 5) And does the 'pseudotime clock' for differentiation of B-cells potentially agree with the expression profiles over time following antigen stimulation? I understand there are limited samples looked at during this snSeq, but either the authors could look at an expanded sample set or at least examine the data they have over time from this study to suggest whether the pseudotime analysis for differentiation of B-cell populations is in agreement with what would be expected in the spleen following antigen challenge over time?

Answer 5) Our data represents a snap-shot of relatively few B cells per fish at a few time points, of which only a few cells are expected to be antigen-specific. Therefore we do not have enough data to conclude. In general, while there was change in gene expression across the B cell sub-clusters, there is no difference in expression pattern of these genes across the infection experiment.

Question 6) Line 413 – 'subsets'

Answer 6) Thank you, we have added this correction.

Question 7) Line 416 – '...is extremely problematic'

Answer 7) We believe the current sentence is correct so have kept it as it is. (Line 547)

REVIEWERS' COMMENTS:

Reviewer #1 (Remarks to the Author):

In the revised version of the manuscript, the authors have addressed all points and questions. I very much appreciate their constructive solutions and support the publication of the manuscript.

Reviewer #2 (Remarks to the Author):

Dear Editor and Authors,

All points raised in my review were covered by the authors responses. It is an interesting well written manuscript and a step forward in the field.

Reviewer #3 (Remarks to the Author):

The authors gave addressed concerns appropriately.